# Mechanism of interaction of an endofungal bacterium *Serratia marcescens* D1 with its host and non-host fungi

**Dibya Jyoti Hazarika**[1], **Trishnamoni Gautom**[1], **Assma Parveen**[1], **Gunajit Goswami**[2], **Madhumita Barooah**[1], **Mahendra Kumar Modi**[1], **Robin Chandra Boro**[1] *

**1** Department of Agricultural Biotechnology, Assam Agricultural University, Jorhat, India, **2** DBT-North East Centre for Agricultural Biotechnology, Assam Agricultural University, Jorhat, India

* robinboro@gmail.com

**Data Availability Statement:** Bacterial 16S rRNA gene sequence and Fungal ITS sequence has been submitted to GenBank (http://www.ncbi.nlm.nih.gov/) (accession numbers: MF893336.1,

## Abstract

Association of bacteria with fungi is a major area of research in infection biology, however, very few strains of bacteria have been reported that can invade and reside within fungal hyphae. Here, we report the characterization of an endofungal bacterium *Serratia marcescens* D1 from *Mucor irregularis* SS7 hyphae. Upon re-inoculation, colonization of the endobacterium *S. marcescens* D1 in the hyphae of *Mucor irregularis* SS7 was demonstrated using stereo microscopy. However, *S. marcescens* D1 failed to invade into the hyphae of the tested Ascomycetes (except *Fusarium oxysporum*) and Basidiomycetes. Remarkably, *Serratia marcescens* D1 could invade and spread over the culture of *F. oxysporum* that resulted in mycelial death. Prodigiosin, the red pigment produced by the *Serratia marcescens* D1, helps the bacterium to invade fungal hyphae as revealed by the increasing permeability in fungal cell membrane. On the other hand, genes encoding the type VI secretion system (T6SS) assembly protein TssJ and an outer membrane associated murein lipoprotein also showed significant up-regulation during the interaction process, suggesting the involvement of T6SS in the invasion process.

## Introduction

Bacterial invasion into eukaryotic cells is one of the major areas of research in infection biology, whereby they employ different strategies to invade the host cells. These interactions are highly complex, and the type of interaction depends on the bacterium, host, as well as environmental factors [1–4]. In the evolution of the eukaryotic cell, the acquisitions of mitochondria and plastids resulting from eubacterial invasions through symbiosis were important events. These acquisitions acted as compartmentalized bioenergetic and biosynthetic factories in the evolved eukaryotic cells [5]. Several studies on bacteria inhabiting eukaryotic organisms including plants, animals, insects as well as nematodes have been reported [6–9]. Endosymbiotic bacteria residing in arbuscular mycorrhizal fungi (Glomeromycota) and their spores were first reported in 1970s as Bacteria-Like Organisms (BLOs) [10]. Later, vertical and horizontal transmission of endofungal bacterium *Burkholderia rhizoxinica* was thoroughly characterized [11,12].

MH536679.1). Strains and experimental data will be available upon request.

**Funding:** This work was supported by the Department of Biotechnology, Ministry of Science and Technology, Government of India under the Project sanction number BT/552/NE/U-Excel/2014 to RCB, but without any influence over the experimental procedures and the results. http://dbtindia.gov.in/ The funder had no role in study design, data collection and analysis, decision to publish, or preparation of the manuscript.

**Competing interests:** The authors have declared that no competing interests exist.

Endobacterial association with fungi governs several physiological processes in the host fungi including sporulation, biomass production, lipid metabolism, etc. [13]. Bacterial-fungal interactions (BFIs) also play crucial role in metabolite production leading to its importance in ecology, agriculture, food processing, and pharmaceutical research. In BFIs, alterations in the cellular behavior may lead to differential expression of genes [2,14], for instance, alteration of secondary metabolite biosynthesis. Some metabolites and toxins synthesized during BFIs confer pathogenicity to the fungal or bacterial entity inhabiting their host. Earlier studies on BFIs revealed that some fungal toxins and metabolites were actually produced by their bacterial partner residing within the fungal hyphae [15,16]. In many cases, various natural products from eukaryotic organisms such as tunicates, sponges, insects, etc. are assumed to be actually synthesized by their associated bacterial endosymbionts [17]. Endofungal bacteria can also govern host sexuality through transcriptional regulation of host receptor genes [18]. Transcriptional changes also occur in the bacterial partner resulting into metabolic and physiological changes needed for their interaction with host. For e.g. *Burkholderia terrae* shows chemotaxis, metabolic activity and oxidative stress responses during its interaction with the host fungus *Lyophyllum* sp. [19]. In a recent study on *Burkholderia rhizoxinica*, involvement of novel pyrrole-substituted depsipeptides (endopyrroles) have been discovered [20]. Heptarhizin, a non-ribosomal cyclopeptide is produced in geographically constrained strains *B. rhizoxinica* under symbiotic conditions [21].

Scarce amount of information is available about the avenues and mechanisms that permit bacterial attachment and invasion into fungal hyphae. Recently, a linear lipopeptide (Holrhizin A) is discovered through genome mining, which helps the endosymbiotic *B. rhizoxinica* during host colonization [22]. Deveau et al. (2018) [4] has reviewed the possible mechanisms employed by endofungal bacterium to invade fungal hyphae and establish physiological association with their host. Involvement of type II secretion system (T2SS) was reported in *Burkholderia* endosymbiont for active invasion of fungal hyphae [23]. Role of type II secretion system (T2SS) in bacterial-fungal interaction has been critically reviewed by Nazir et al., (2017) [24].

In this study, an attempt was made to identify and characterize a novel bacterial-fungal interaction and the related molecular mechanism that permits the invasion and colonization of the endofungal bacterium in their host fungi. Here we report the endofungal nature of *Serratia marcescens* in a soil borne fungus *Mucor irregularis*. Various strains of *Serratia marcescens* have been isolated from rhizospheric soil [25,26] and from several plants as endophytes [27–29]. But the endofungal lifestyle of this bacterial species has not been reported earlier. However, *Serratia* is reported to show antagonistic activity against bacteria and fungi, and some of them feed on fungi by biofilm formation around the fungal hyphae leading to hyphal death [30,31]. Comparative genome analysis of *Serratia marcescens* has demonstrated that most of the members of this species posses type VI secretion system encoding gene clusters within their genomes [32]. *S. marcescens* utilizes T6SS to compete among bacterial competitors by secreting virulent factors [33]. Apart from T6SS, the T2SS has also been detected in *S. marcescens*, which secretes chitinases out of the cell [34]. This study aimed to elucidate the the involvement of secretion systems and secondary metabolites of *Serratia marcescens* during its interaction with its fungal host.

## Materials and methods

### Bacterial screening in the fungal cultures

A total of 65 fungal isolates were screened for the presence of possible endofungal bacteria. Fungal cultures were isolated from diseased leaves of rice, sugarcane, tea and horticultural crops as well as from rhizospheric soil. For isolation from the leaves and other plant tissues,

the plant parts with symptoms of disease were washed with clean water and cut into small pieces with sterile surgical blade (Himedia, India). Small pieces were then surface sterilized with 1% sodium hypochlorite (Himedia, India) solution for 2 min followed by three repeated washing with sterile distilled water to remove the surface contaminants. Surface sterilized samples were then inoculated on potato dextrose agar (Himedia, India) medium containing 4 g/l potato extract, 20 g/l dextrose and 15 g/l agar. For isolation of fungi from rhizospheric soil, the soil plate method [35] was used. Plates were incubated at 28 ˚C for 2–4 days and observed thereafter for presence of different fungal colonies. Individual colonies were used for screening of endofungal bacteria.

Initial screening was carried out by visual observation of slimy bacterial appearance in the fungal colonies when grown on PDA supplemented with 5 g/l peptone and 5 g/l sodium chloride. Fungal cultures with possible association of bacteria were subjected to PCR based screening for the confirmation of bacterial presence in the fungal cultures directly as well as after two subsequent subculturing in PDA medium.

The hyphae from the fungal cultures were taken in a 2 ml vial and washed repeatedly with 0.85% NaCl solution to remove any surface contaminants of the hyphae. Total genomic DNA from the fungal hyphae were isolated using HiPurA™ Soil DNA Purification Kit (Himedia, India; the protocol is available online at www.himedialabs.com). The fungal metagenome was further amplified with bacterial 16S rRNA gene specific universal primers: 16S-F (5'-AGAGT TGATCCTGGCTCAG-3') and 16S-R (5'-ACGGCTACCTTGTTACGACTT-3'). Amplified products of ~1500 bp were considered as positive for the presence of bacteria [15].

## Microscopic detection of the endofungal bacteria

Fungal hyphae stained with lactophenol cotton blue as well as unstained hyphae were visualized under 100x magnification of Olympus BX51 light microscope (Olympus Corporations, Japan). The fungal hyphae were also stained with a mixture of SYTO9® and propidium iodide from LIVE/DEAD BacLight bacterial viability kit (Life Technologies, USA), which can differentiate between live and dead bacterial cells. Hyphae were observed under 100x magnification of Olympus BX51 fluorescent microscope (Olympus Corporations, Japan) at an excitation wavelength of 480 nm. The fungal culture grown on PDA plate containing 75 mg/l of streptomycin was taken as negative control in both the cases.

## Isolation of the endofungal bacteria

The fungal hyphae were crushed in sterile saline solution and centrifuged at 2500 rpm for 1 min. The supernatant was carefully separated and streaked onto nutrient agar (NA; Himedia) plates containing 5 g/l peptone, 5 g/l NaCl, 1.5 g/l beef extract, 1.5 g/l yeast extract and 15 g/l agar. The bacterial colonies thus obtained were again streaked onto fresh NA plates. To obtain bacteria free culture of the fungal isolate, the fungal isolate was grown on PDA medium supplemented with 75 mg/l of streptomycin. The bacterial and fungal pure cultures were maintained separately which were used as control cultures in further experiments.

## Characterization of the endofungal bacteria

The bacteria isolated from the fungal culture SS7 living as an endobacterium was selected for further characterization. Preliminary characterization included colony morphology, Gram's staining and biochemical characteristics of the bacterial isolate.

## Bacterial identification by Fatty Acid Methyl Ester (FAME) analysis

Total cellular fatty acids of the bacterial isolate were profiled using the MIDI Sherlock Microbial Identification System (Microbial ID Inc., USA). Instant FAME Method kit was used for Fatty acid extraction and methyl ester generation according to the manufacturer's instruction. FAME analysis was performed on a Gas chromatography (GC) system (Agilent 6890N analyzer, Agilent Technologies, USA) using calibration standards (#1300-AA; MIDI, Inc.). Sherlock® Microbial Identification (version 4.5) software was used to analyze individual fatty acids in the GC peaks. The bacterial identification was carried out by comparing fatty acid profile with the database of Instant Environmental TSA library (version 1.10). Results with similarity index (SI) $\geq$ 0.5 were considered as acceptable for FAME identification [36].

## Molecular identification of the bacterial and fungal isolates

Molecular identification of the bacterial and fungal isolates was carried out by sequencing the 16S rRNA gene and internal transcribed spacer (ITS) region of the bacterial and fungal genome respectively. Genomic DNA was extracted from both the bacterial and fungal isolates. Bacterial 16S rRNA gene was amplified using universal primers: 27F (5'–AGAGTTTGATCC TGGCTCAG–3') and 1492R (5'–GGTTACCTTGTTACGACTT–3'). To confirm the fungal identity, the ITS region of the fungal genome was amplified with a pair of universal primers: ITS1 (5'–TCCGTAGGTGAACCTGCGG–3') and ITS4 (5'–TCCTCCGCTTATTGATATGC– 3') [37]. The amplified PCR product was resolved on a 1.2% agarose gel to observe the integrity of the amplified product. The PCR products were purified using GenElute™ PCR Clean-Up Kit (Sigma-Aldrich, USA) followed by cloning of the same using pGEM-T Easy Vector System I. Plasmids from the positive clones were sequenced through external vendor (Bioserve Biotechnologies, India) in an ABI 3130 automated DNA Sequencer (Applied Biosystems, USA) with vector specific primers. The 16S rRNA gene and ITS region sequencing reads were manually aligned using MEGA 6.0 software [38] and the sequence thus obtained was compared with GenBank database of NCBI (Bethesda, MD, USA) using BLAST.

## Phylogenetic analysis

Phylogenetic analysis of the bacterial and fungal isolates was carried out. Reference bacterial 16S rRNA gene sequences and fungal internal transcribed spacer (ITS) sequences were retrieved from GenBank database of NCBI (Bethesda, MD, USA). The 16S and ITS sequences were aligned separately with *S. marcescens* D1 16S rRNA gene sequence and *M. irregularis* SS7 ITS sequence, respectively using ClustalW 1.6 in MEGA 6.0 [38]. The phylogenetic trees were constructed using Maximum Likelihood method [39]. Initial tree(s) for the heuristic search were generated by applying the Neighbor-Joining method to a matrix of pairwise distances estimated using the Maximum Composite Likelihood (MCL) approach. The evolutionary history was inferred based on Tamura-Nei model [40].

## Interaction of *S. marcescens* with different fungal species

Ascomycota/Basidiomycota/Zygomycota and *S. marcescens* D1 strains were inoculated as spots on NA+PDA medium (containing 2.8% NA and 4.9% PDA) and incubated at 28 ˚C for 2–5 days. The fungal mycelial plugs were inoculated on the center of the culture plates and allowed to grow for at least 24 h. The bacterial culture was spotted afterwards on four adjacent sides of the growing fungal colony and incubated at 28 ˚C. Bacterial spreading along the mycelium was assessed visually from the next day of bacterial inoculation. Microscopic observation was also carried out to detect bacterial cells within the growing fungal hyphae. A bacteria free

culture of each fungal strain was used as control. Bacterial cells within the fungal hyphae was visualized by staining the hyphae with SYTO9® (Life Technologies, USA) and fluorescence was detected using a Leica confocal laser scanning microscope.

## Viability assessment of bacteria containing fungal mycelium

A 5 mm disc of pre-grown fungal culture was inoculated in NA+PDA medium and allowed to grow overnight at 28 ˚C. The bacteria was inoculated on the next day adjacent to the growing edges of the fungal colony. The plates were again incubated at 28 ˚C allowing the growth of the fungal colony over the bacterial colonies. The spreading of the bacteria along the fungal hyphae was visualized periodically till 7 days. At each time point, the hyphal plugs from the bacteria containing regions were picked with a sterile toothpick, placed inverted on the surface of a PDA plate with or without antibiotic for 48 h at 28 ˚C, and scored for radial growth of the fungus. Mycelial plug from a fungal culture plate lacking the bacteria was also inoculated and scored as positive control. No detectable fungal growth was defined as 100% killing by the bacteria, while significant fungal growth was defined as the viability of the fungal hyphae.

## Bacterial movement in the aerial fungal hyphae

Solid medium containing 0.6% potato dextrose broth (PDB), 0.4% nutrient broth (NB) and 1.5% agar was used to conduct this experiment. A 5 mm width of the agar was aseptically expunged through the diameter of the plate and the fungal culture was inoculated on one side of the agar at a distance of 3 cm from the expunged region. The bacteria was inoculated on the other side, very adjacent to the expunged region. The plates were incubated at 28 ˚C for 48 to 96 h to allow bridging of the gap by the fungal hyphae. Bacterial movement was confirmed visually as well as by stereo microscopy. Sampling was carried out from the bacteria containing fungal mycelium with a sterile toothpick and these samples were point inoculated on NA plates and grown for 24 h at 28 ˚C to detect the viable bacterial cells.

## Quantification of sporulation in *M. irregularis* during interaction with bacteria

*Mucor irregularis* cultures were grown in PDA medium in presence or absence of *Serratia marcescence* D1. Effect of prodigiosin, the red pigment produced by the bacterium, was also studied to unlock its role on spore formation. Prodigiosin was extracted from the bacterial cells (pregrown for 48 h on NA plates at 28˚C) using 1 mM hydrochloric acid: acetone: ethyl acetate (1: 2: 3). The upper pink-red organic phase was separated after 4 h, concentrated to dryness under reduced pressure and re-dissolved in chloroform. Prodigiosin was further purified using silica gel thin layer chromatography (TLC) and identity of the purified prodigiosin was confirmed using Liquid Chromatography-Mass Spectrometry (unpublished work). Role of prodigiosin on fungal sporulation was tested by growing the fungal strain on PDA medium embedded with 100 µg prodigiosin. Bacteria treated, prodigiosin treated and control cultures were incubated at 28 ˚C. After each 24 h of incubation, a 1 cm$^2$ of fungal mycelium was recovered from the sub-terminal portion of the fungal colony in each plate. The mycelia were then transferred to a 2 ml vial containing 1 ml saline solution. The tube was vortexed for 5 min and 100 µl of the suspension was mixed with 100 µl of lactophenol cotton blue stain (Himedia, India). Spore count was carried out using a haemocytometer under 100X objective of Olympus BX51 microscope (Olympus Corporation, Japan). The experiment was repeated at least three times.

## Evaluation of the effect of prodigiosin on the fungal cell membrane

Mycelial plug from a pre-grown culture of *M. irregularis* was inoculated on PDA plate and incubated at 28°C for 48 h. After 24 h of inoculation, small amount of hyphae were harvested and taken in a 2 ml vial containing 0.1 ml of saline solution. Hyphal suspension were then treated with prodigiosin to a final concentration of 0, 500 and 1000 μg/ml prodigiosin solubilized in DMSO. Tubes were then incubated for 1 h at room temperature. After incubation, propidium iodide (Life Technologies, USA) was added to each tube to a final concentration of 1 mM and incubated for 15 min. The hyphae were then taken in a clean slide and observed under Leica confocal laser scanning microscope.

## Primer designing for quantitative real-time PCR

Five genes from prodigiosin biosynthetic gene clusters were selected to study the transcriptomic changes in prodigiosin biosynthesis during the interaction of *S. marcescens* D1 and *M. irregularis* SS7. Primers were designed towards sequences of five *S. marcescens* biosynthetic genes to amplify target sequences ranging from 130–180 bp using NCBI primer BLAST [41]. Primers were also designed for two genes that encode type VI secretion system (T6SS) proteins TssJ and murLP, a gene (*chiA*) encoding the fungal cell wall degrading protein chitinase A, and two constitutively expressed genes *rpoD* and *gyrB* encoding the RNA polymerase sigma factor RpoD and DNA gyrase subunit B, respectively (S1 Table).

## Quantitative real-time PCR

Total RNA was extracted from *Serratia marcescens* D1 pure culture as well as *Serratia-Mucor* interaction culture using Trizol method. The first strand cDNA synthesis was carried out with 200 ng of total RNA in a 20 μl reaction using GoScript™ Reverse Transcription System (Promega Corporation, USA). Expression profiles of the target genes were quantified by quantitative real-time PCR (qRT-PCR) in a QuantStudio 5 Real-Time PCR System (Applied Biosystems, USA). Each 20 μl reaction contained 1 μl of 2-fold diluted cDNA template, 1 μl (200 nM) of each primer and 10 μl of GoTaq® qPCR Mastermix 2X (Promega Corporation, USA). The amplification program consisted of: initial holding at 50 °C for 2 min, denaturation at 95 °C for 10 min, 40 cycles of 95 °C for 15 s and 60 °C for 1 min. Melt curve analysis was carried out to ensure single amplification product. Expression of the target genes was normalized by *rpoD* gene and constitutive expression of *gyrB* gene was also tested.

# Results

## Screening and detection of endofungal bacteria

The preliminary screening and PCR amplification of the fungal metagenomes for 16S rRNA gene indicated the presence of bacterial gene in seven fungal DNA samples: SS7, OR4.1, AAU-R4, AAU-R6, SC2.2, SC4.6, HB8 (S1 Fig). However, subsequent subcultures from the colony edges of these fungal isolates on PDA medium eliminated the bacterial appearance from the fungal colonies except for SS7. The fungal culture SS7 was detected with a pink-red pigmentation (Fig 1h). This pink-red pigmentation persisted after sub-culturing on PDA medium followed by sub-culturing in NaCl and peptone containing PDA medium. The PCR amplification of 16S rRNA gene from the isolated DNA after two subsequent subcultures also indicated the presence of bacterial DNA in the fungal metagenome of SS7, but not in that of the other fungal isolates.

The above experiment confirms the presence of bacteria in the fungal culture SS7, which was further studied using light microscopy and fluorescence microscopy to confirm their

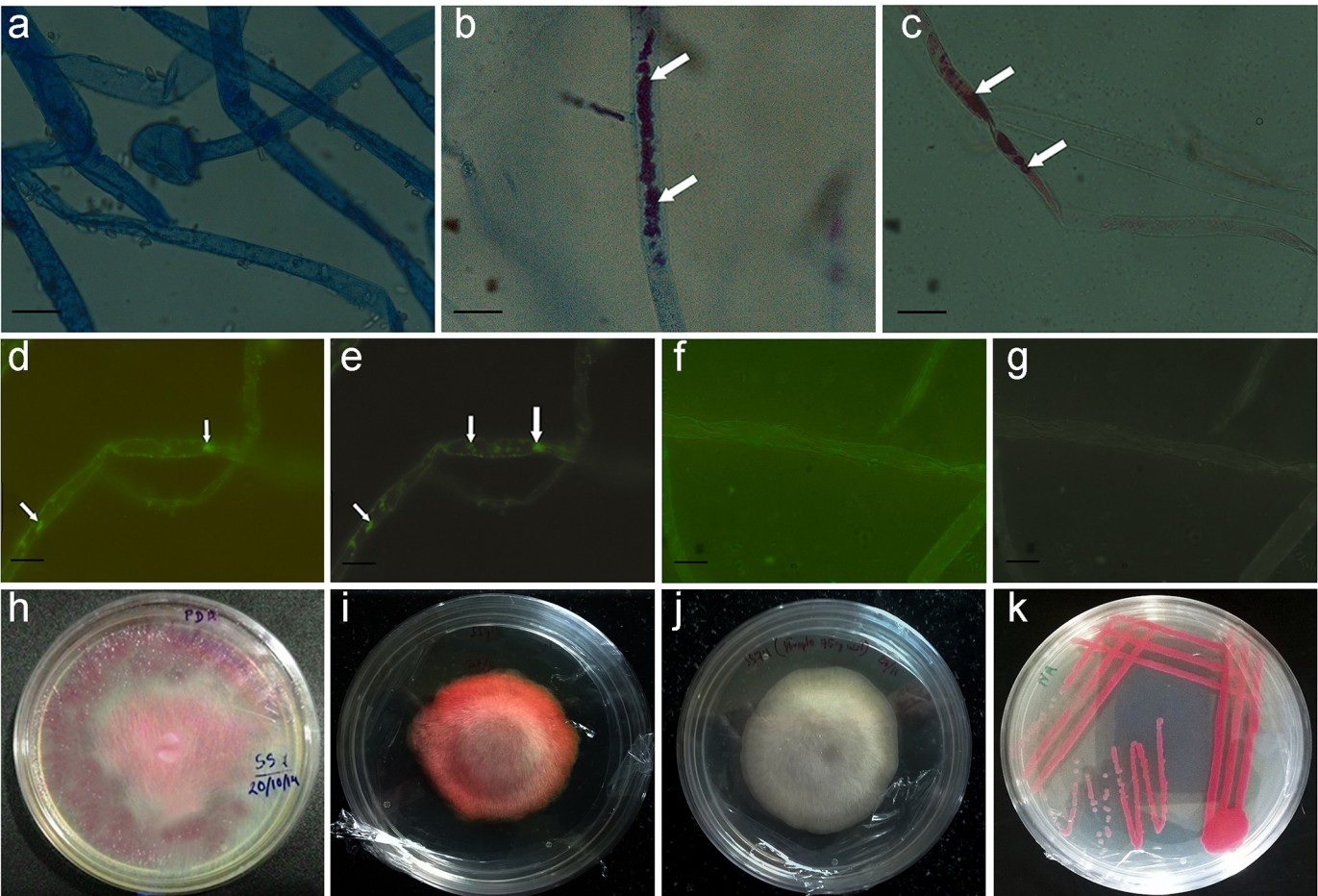

**Fig 1. Observation of fungal hyphae of hyphae of fungal sample SS7 under light microscopy and fluorescent microscopy (1a-1g).** Isolated fungal and bacterial cultures on solid media (1h-1k). **a**. Fungal hyphae from control culture stained with lactophenol cotton blue; **b**. fungal hyphae from fungal culture containing bacteria stained with lactophenol cotton blue; **c**. unstained fungal hyphae containing bacteria; **d**. fluorescence-bright field microscopy combined image and **e**. fluorescence microscopic (without bright field) image showing bacterial cells within fungal hyphae; **f**. fluorescence-bright field microscopy combined image and **g**. fluorescence microscopic (without bright field) image showing no bacterial cells inside control hyphae from antibiotic containing medium; hyphae in **d.–f**. were stained with SYTO9®; **h**. The mother culture of SS7 on peptone and NaCl containing PDA medium; **i**. SS7 subcultured from the colony edge of the mother culture; **j**. fungal pure-culture on streptomycin containing PDA medium; **k**. isolated bacterial pure-culture on NA medium.

endofungal localization. The light microscopic images detected the presence of pink-red pigmentation within the fungal hyphae of SS7 (Fig 1b and 1c). No pink-red pigmentation was observed in the control hyphae of SS7 grown in the PDA medium containing 75 mg/l streptomycin (Fig 1a). This result suggested that the pink-red pigmentation was due to the bacterial cells residing within the fungal hyphae. The fluorescence microscopy clearly revealed the presence of the endobacterium within the hyphae. The fluorescent dye SYTO9® detected live bacterial cells within the fungal hyphae of SS7 and were observed as green fluorescence (Fig 1d & 1e). No fluorescence was detected in the hyphae from the control culture of SS7 (Fig 1f and 1g).

## Identification of the bacterial isolate

The bacterium isolated from SS7 was designated as D1 in the further experiments. The co-cultures and pure cultures of SS7 and D1 are shown in Fig 1h–1k. The preliminary morphological and biochemical tests were carried out and results suggested that the bacterium was Gram

negative and produced a pink-red pigment in NA and tryptic soy agar (S2 Table). The bacterial pigment production was reduced in PDA, but pigment production was revived when PDA was supplemented with peptone. This result suggested that some of the amino acids may be crucial for pigment production which are absent in PDA medium.

The bacterial isolate D1 was tested against 17 standard antibiotics in Mueller Hinton Agar plates using standard antibiotic disks (Himedia Laboratories, India). It was observed that the bacterium was resistant to amoxyclav, ampicillin, cephalothin, oxacillin, penicillin-G and sulphatriad. However, bacterial pigment production was completely inhibited by sulphatriad within its zone of action. The bacterial isolate was found to be susceptible to cefoxitin, ceftazidine, chloramphenicol, clindamycin, erythromycin, gentamycin, ofloxacin, streptomycin, teicoplanin, tetracycline and vancomycin (S3 Table).

The FAME profile of the bacterial isolate matched with that of *Serratia marcescens* showing a similarity index of 0.504 with *Serratia marcescens* GC-subgroup A. The bacterial identity was further confirmed as *Serratia marcescens* D1 based on the BLAST percent identity score of 16S rRNA gene sequence. The sequence showed 96.04% similarity to its closest strain *S. marcescens* AG2102. A phylogenetic tree was constructed to study the evolutionary relationship of *S. marcescens* D1 with different species. *Serratia marcescens* D1 was grouped with two other reference strains of *S. marcescens* followed by other species of *Serratia*. The tree suggested its closest relationship to *S. marcescens* AG2102 with an evolutionary distance of 0.024 (Fig 2a). Finally, the 16S rRNA sequence of *S. marcescens* D1 was submitted to GenBank (Accession No. MF893336.1).

## Identification of the fungal isolate

The fungal isolate SS7 was identified based on the morphological and molecular characteristics. The colony morphology of SS7 pure culture on the PDA medium was whitish and woolly in appearance in the obverse side, while reverse side was yellowish. Well-developed rhizoids like structures and spherical sporangia were detected in the microscopic observations. Sporangiospores were irregular in shape and size. The morphological characteristics indicated the identity of the fungal isolate to be *Mucor irregularis* (synonym. *Rhizomucor variabilis*) [42] which was further confirmed by the ITS sequence similarity in BLAST. The BLAST analysis showed 96.07% similarity of the sequence to its closest member of the species (*Rhizomucor variabilis* CBS 103.93). The phylogenetic analysis also suggested its closest relationship to *R. variabilis* CBS 103.93 with an evolutionary distance of 0.029 (Fig 2b). The isolate SS7 was clustered together with *Mucor irregularis* and *R. variabilis* strains followed by other members of the genus *Mucor*. The sequence of *Mucor irregularis* SS7 was submitted to GenBank (Accession No. MH536679.1).

## Interaction of *S. marcescens* with different fungal species

We used five Ascomycota strains *viz. Aspergillus flavus*, *Aspergillus nomius*, *Penicillum citrinum*, *Fusarium solani* and *Fusarium oxysporum* and six Basidiomycota strains *viz. Pleurotus ostreatus*, *Sarcodon* sp., *Chlorophylum molybdites*, *Pycnoporus coccineus*, *Coprinellus* sp. and *Leucocoprinus* sp., to assess the spreading of *Serratia marcescens* over fungal mycelia. *Mucor irregularis* was used as positive control for assessment of bacterial spreading. The results suggested that the bacteria was able to spread in the culture of *Mucor irregularis* SS7 and *Fusarium oxysporum* SC7.1. Bacterial spreading was very low in *Fusarium solani* hyphae, while, for the rest of fungal strains *viz. Aspergillus flavus*, *Aspergillus nomius*, *Penicillium citrinum*, *Pleurotus ostreatus*, *Sarcodon sp.*, *Chlorophyllum molybdites*, *Pycnoporus coccineus*, *Coprinellus sp*. and

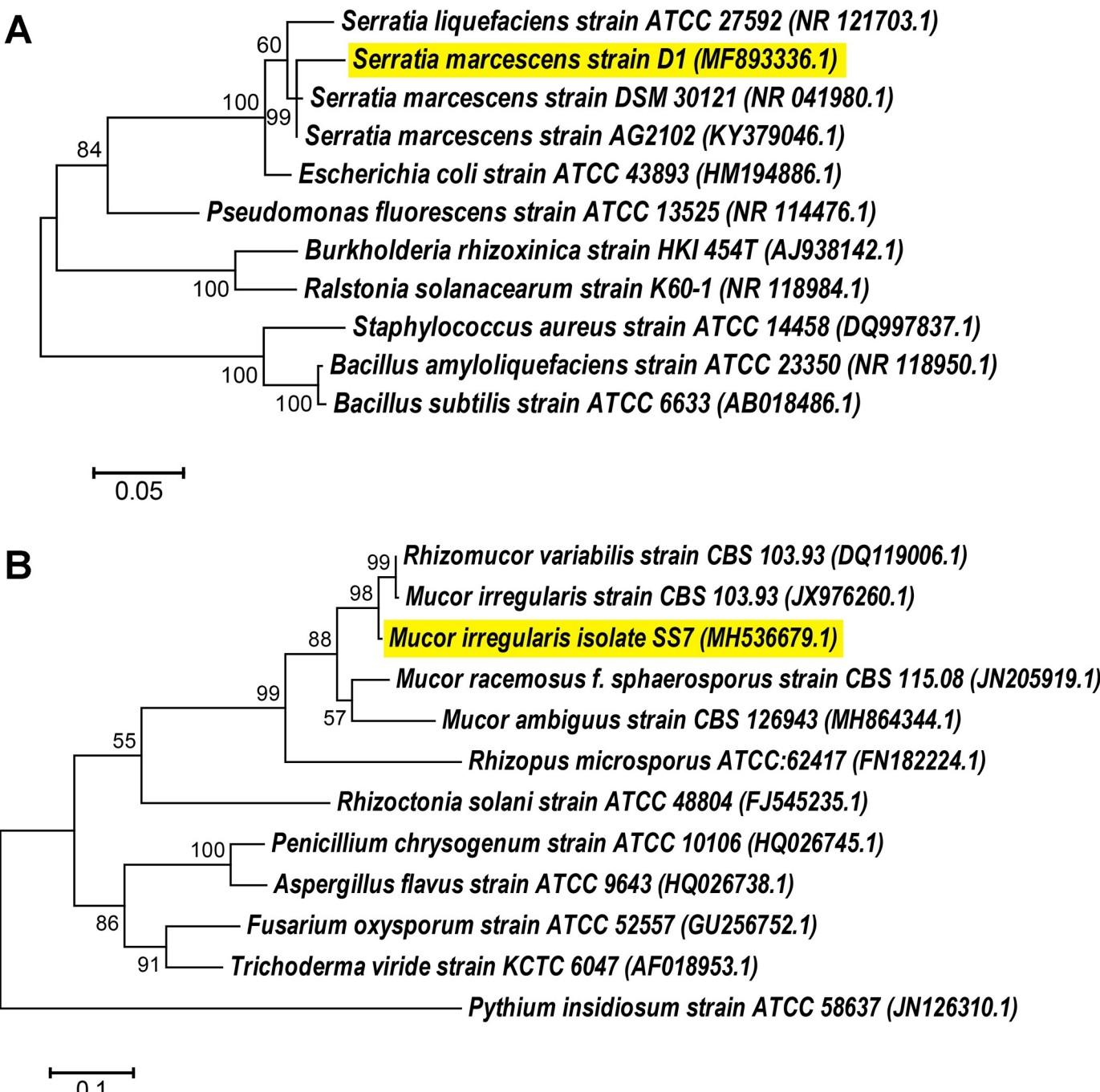

**Fig 2. Phylogenetic relationships of the bacterial and fungal isolates with other strains of bacteria and fungi, respectively.** Both the trees were constructed using Maximum Likelihood method in MEGA6. **a**. Phylogenetic tree of the 16S rRNA gene showing relationship of *Serratia marcescens* D1 with other isolates; **b**. Phylogenetic relationship of the internal transcribed spacer region showing relationship of *Mucor irregularis* SS7 with other strains. The reference sequences were retrieved from GenBank and accession numbers are given within brackets.

*Leucocoprinus sp.*, no significant spreading of the bacteria was detected inside the fungal hyphae (S4 and S5 Figs).

Fluorescent microscopic images of two Ascomycete fungi *Aspergillus flavus* and *A. nomius* and Zygomycete *M. irregularis* were shown in Fig 3. Green fluorescence was only detected in

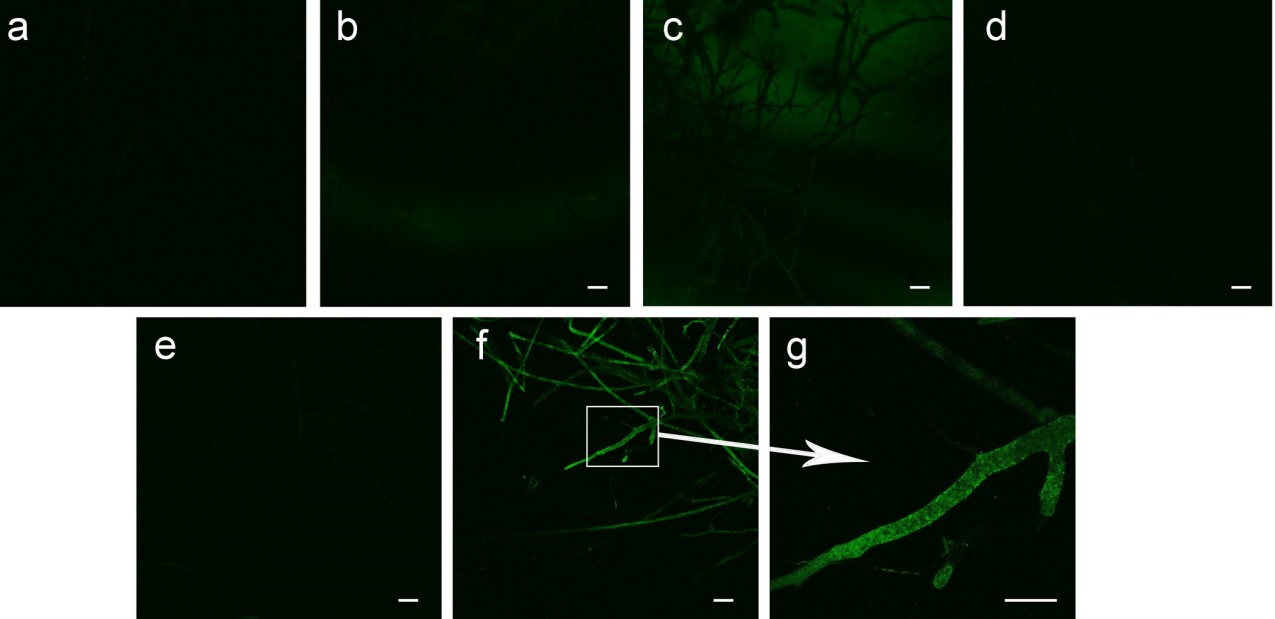

**Fig 3. Fluorescence based detection of bacterial presence in the fungal hyphae using SYTO9® fluorescent dye. a**. hyphae from *Aspergillus flavus* control culture; **b**. hyphae from *A. flavus* culture treated with *S. marcescens*; **c**. hyphae from *A. nomius* control culture; **d**. hyphae from *A. nomius* culture treated with *S. marcescens*; **e**. hyphae from *M. irregularis* control culture; **f**. hyphae from *M. irregularis* culture treated with *S. marcescens*; **g**. Magnified image of bacteria containing *M. irregularis* hyphae. Scale bar: 10 μm.

*M. irregularis* hyphae indicating the presence of *S. marcescens* cells within the hyphae, while no bacteria was detected inside the tested Ascomycete hyphae.

## Viability assessment of bacteria containing fungal mycelium

We noticed that the bacteria containing fungal culture of *Mucor irregularis* SS7 grew after re-inoculation on PDA medium. Bacterial presence in the re-inoculated fungal culture was also observed with red pigmentation. However, when the mycelial plug from the co-culture was inoculated on PDA medium containing streptomycin, the bacterial presence was not detected and the normal fungal growth was observed indicating that *Serratia marcescens* D1 spread over and in the mycelia of *Mucor irregularis* SS7 without killing the fungus (Fig 4). The mycelial plugs remained viable till 7 days of interaction.

In case of *Fusarium oxysporum* SC7.1, the bacteria containing mycelial plugs taken after 2–7 days of interaction failed to regrow in PDA medium after 48 h of re-inoculation. Due to excessive bacterial growth on the re-inoculated mycelial plug, no fungal growth was observed. Use of antibiotic containing plate eliminated the bacterial growth on mycelial plug, but the mycelial plug still failed to germinate indicating the loss of viability of the fungal mycelium.

## Migration of *S. marcescens* through the aerial fungal hyphae

The removal of agar medium through the diameter of the plate restricted the direct contact of the bacterial cells with the fungal colony. This also restricted the chance of bacterial movement towards the fungal colony. The movement of bacteria in and over the fungal hyphae was clearly observed under stereo microscope at different time intervals (Fig 5) suggesting that the

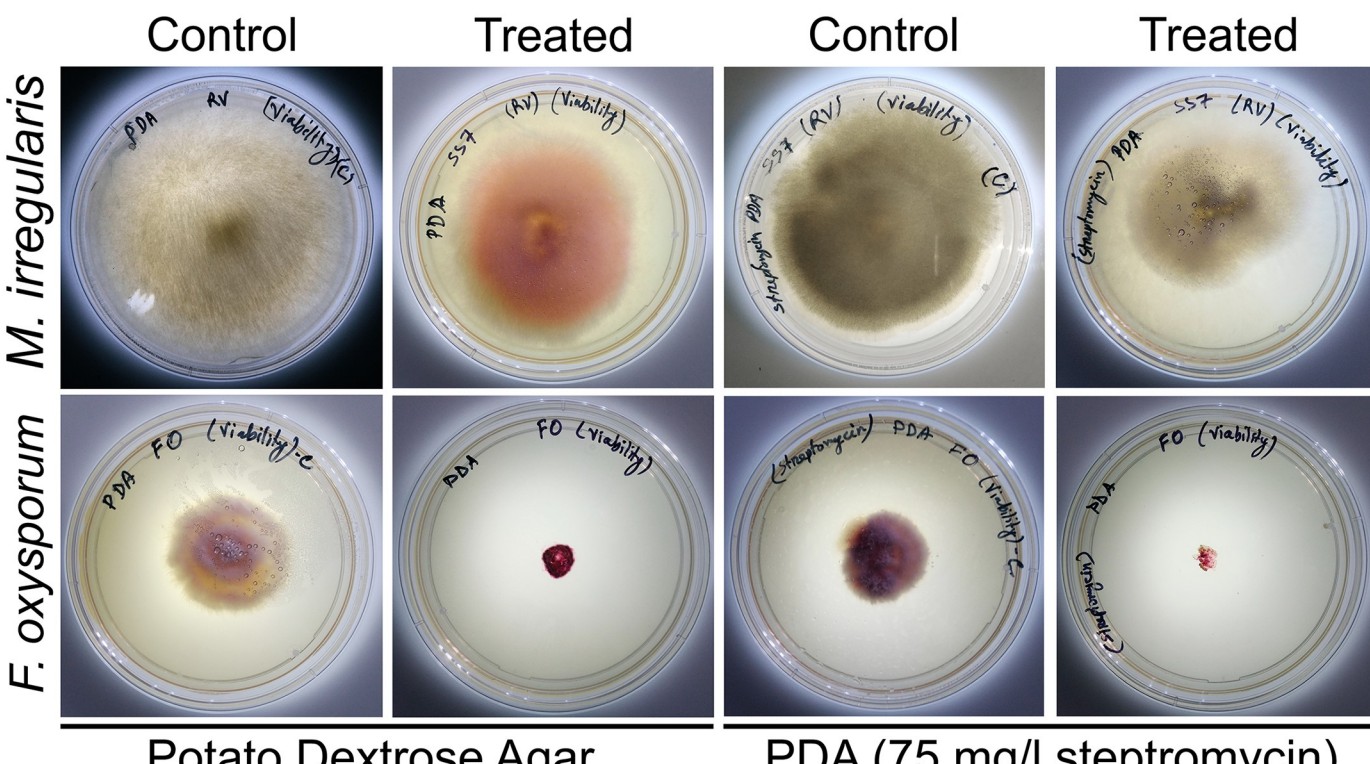

**Fig 4. Viability assessment of *Mucor irregularis* SS7 and *Fusarium oxysporum* SC7.1 hyphae inoculated from bacteria containing cultures.** Mycelial plugs were taken from control and bacteria treated cultures after 7 days of inoculation and re-inoculated on fresh plates with or without streptomycin. Photographs were taken after 48 h of re-inoculation.

bacteria could spread through the fungal mycelium of *M. irregularis*. Migration of bacteria was clearly observed through the aerial fungal hyphae.

## Quantification of sporulation in *M. irregularis* during interaction with bacteria

During the first 24 h of fungal inoculation there was no significant difference in sporulation in the cultures grown in presence or absence of the bacterium. Sporulation in the prodigiosin treated plates was also similar to the other two cultures till 24 h. However, after 48 h till 96 h, *Mucor irregularis* cultures grown in presence of *Serratia marcescence* D1 showed significant reduction in the sporulation compared to the control culture of the fungus. Prodigiosin treated plates showed similar spore count as the control culture after 48 h and 96 h, but higher spore count was recorded after 72 h in prodigiosin treated plate as compared to the control plate (S6 Fig).

## Inhibitory activity of prodigiosin is achieved by increased membrane permiability

Prodigiosin showed potent inhibitory activity against fungal strains in PDA plates. Growth of the fungal strains in the prodigiosin containing medium decreased compared to the control cultures, which could be observed by the reduction in colony diameter. Among the tested fungal strains, *Fusarium oxysporum* was highly suppressed by the treatment of prodigiosin. For all other tested organisms, inhibition of growth was significantly reduced to the same range (S4 Table).

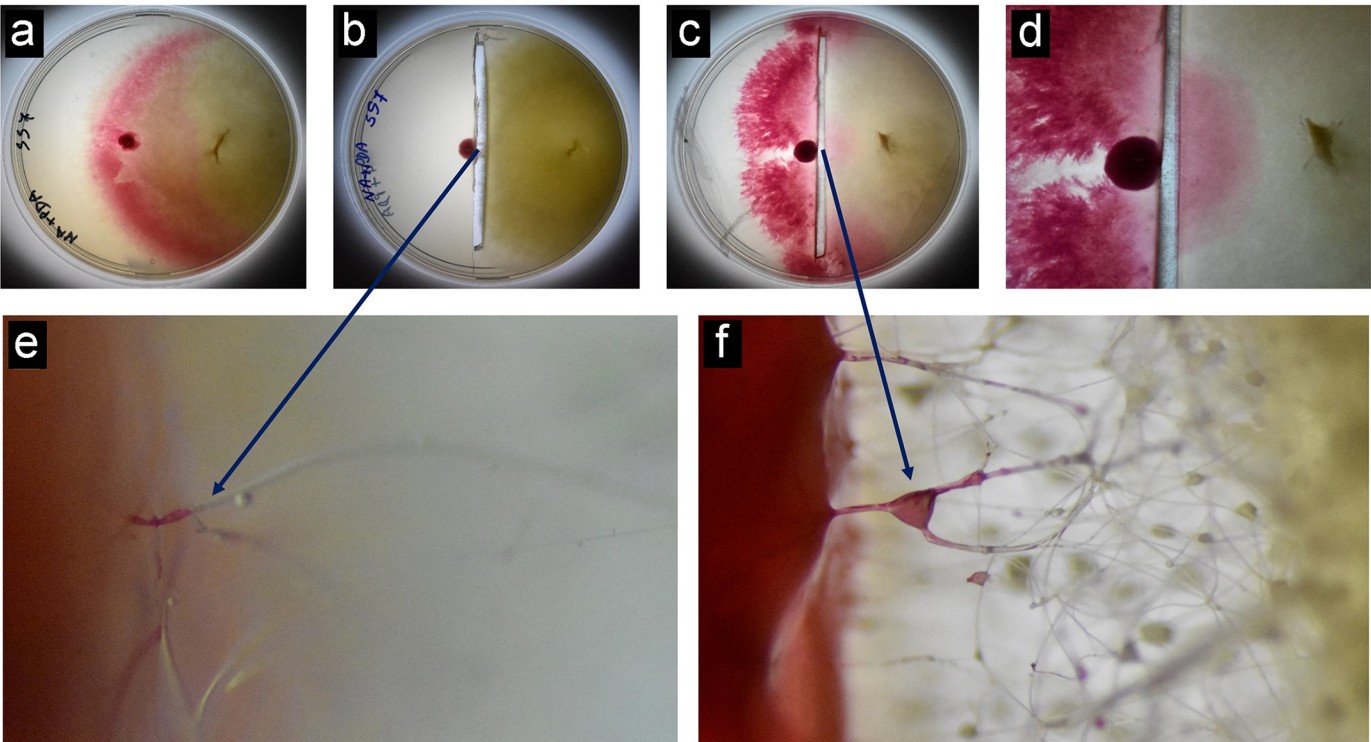

**Fig 5. Photographs showing movement of bacteria inside the aerial hyphae of *M. irregularis* SS7. a**. Bacterial fungal co-culture on solid medium showing bacterial spreading over the in the fungal culture. **b**. Bacterial fungal co-culture (after 48 h of inoculation) separated from each other by removing the culture medium along the diameter of the plate. **c**. Bacterial fungal co-culture (agar removed) after 72 h of inoculation. **d**. Closer view of the plate shown in **c**; **e**. Stereo microscopic observation of the hyphae showing bacterial migration after 48 h of inoculation; **f**. Stereo microscopic observation of the hyphae showing bacterial migration after 72 h of inoculation.

Prodigiosin was able to increase the permeability of *Mucor irregularis* cell membrane. When compared with control, the intensity of red fluorescence increased with increasing concentration of prodigiosin. Due to the increase in cell membrane permeability, more amount of propidium iodide could enter into the hyphae that resulted into increase in the red fluorescence (S7 Fig).

## Mechanism of interaction between *Serratia marcescens* and *Mucor irregularis*

Quantitative real-time PCR analysis of prodigiosin biosynthetic genes showed no differential expression in the two tested conditions *i.e.* bacterial pure culture and bacterial fungal interaction culture. This result suggested that prodigiosin production in *Serratia marcescens* D1 is not dependent on its interaction with fungal host. Chitinase A expression was also not upregulated during interaction. These results clearly indicated that Chitinase A was not involved in the invasion process. Expression levels of other chitinases of *S. marcescens* (Chitinase B, Chitinase C, Chitinase X, etc.) were not tested in this study. On the other hand a significant upregulation of the T6SS associated protein TssJ indicated the involvement of T6SS in the invasion process (Fig 6).

## Discussion

Bacteria have immense potential to inhabit remarkably diverse ecological niches, and in many cases, they form strong mutualistic associations with other organisms. Bacteria often live and reproduce within their host and some of them are vertically transmitted in analogy to

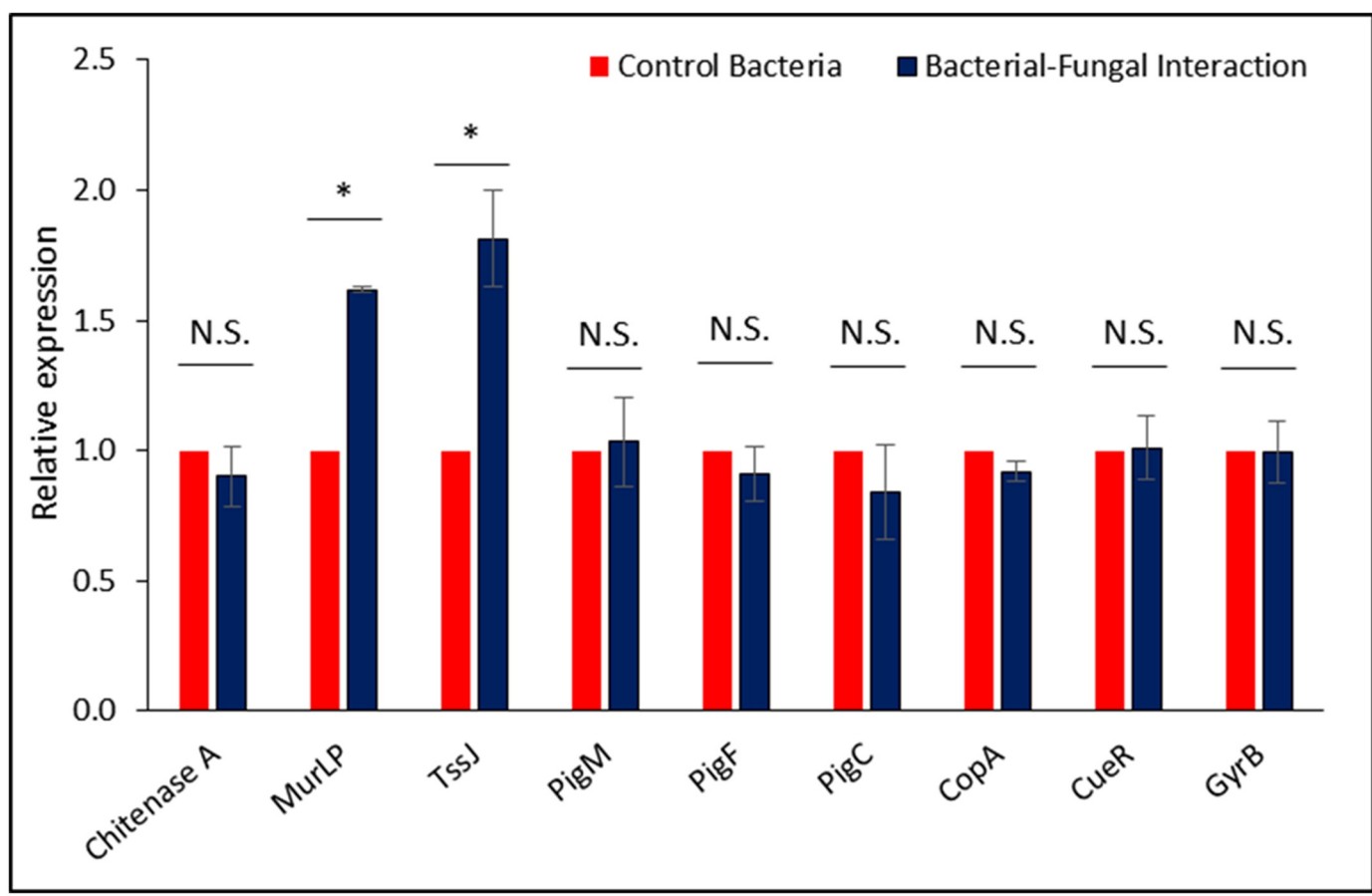

**Fig 6. Relative expression pattern of tested genes hypothesized to be involved in the invasion of fungal hyphae by the endobacterium.** RpoD expression was used for normalization of the qRT-PCR results. GyrB expression was tested for validation of the constitutive expression. Significance was calculated using student's *t*-test with p ≤0.05.

mitochondria. Endosymbiotic bacteria have already been found in marine animals [6], plants (root nodules, leaf galls, stem galls, seeds) [7], insects [8] and worms [9]. Recent studies have reported fungi as a host for bacterial endosymbionts [15,16,43,44]. The rice seedling blight fungus, *Rhizopus microsporus*, and its endosymbiont bacterium, *Burkholderia rhizoxinica* represent a particularly noteworthy example of a bacterial-fungal endosymbiosis [15,16]. The fungus harbours bacterial endosymbionts of the genus *Burkholderia*, which reside within the fungal cytosol. It was reported that the production of rhizotoxin, a potent cell-cycle inhibitor secreted by the plant pathogenic fungus *Rhizopus microsporus*, is dependent upon a bacterial endosymbiont. Strikingly, *Burkholderia* produces rhizotoxin when grown in laboratory media, establishing the endosymbiont as the actual source of this key virulence factor [15]. Another report revealed that endobacteria isolated from the mycorrhizal fungus *Rhizobium radiobacter*, exhibit the same growth promoting effects and induce systemic resistance to plant pathogenic fungi in the same way that the fungus harboring the endobacteria does. Thus, it was proposed that the beneficial effects for the plant result directly from the presence of bacteria [44].

In our study, various fungal isolates have been screened for the presence of endofungal bacteria. We found an endofungal isolate *Serratia marcescens* D1 residing within the hyphae of *Mucor irregularis* SS7. Microscopic observations revealed the presence of the bacteria inside the fungal hyphae. Since the bacteria produced pink-red pigment, the bacterial growth inside

the fungal hyphae could be clearly observed under light microscope. To check whether the bacterial cells or only pigments were taken up by the hyphae, fluorescence microscopic examinations have been performed using SYTO9$^®$ and propidium iodide. Our observations re-established the use of these two fluorescent dyes to detect endofungal bacteria. The dye SYTO9$^®$ can penetrate into bacterial cells and bind with the bacterial DNA to emit green fluorescence under blue light excitations. Propidium iodide can also bind to bacterial DNA but cannot enter live bacterial cells due to impermeability to intact cell walls and therefore only dead cells fluoresce red under green light excitation. Previously, Partida-Martinez and Hertweck (2005) [15] reported the use of this fluorescent dye mixture to detect *Burkholderia rhizoxina* cells within the fungal hyphae of *Rhizopus microsporus*.

Various strains of *Serratia marcescens*, a Gram negative bacteria belonging to the family Yersiniaceae, have been isolated from rhizospheric soil [25,26] and also have been reported as endophytes in different plants [27–29]. However, *Serratia* sp. has been reported to show antagonistic activity against fungi and some of them feed on fungi by forming biofilm around the fungal hyphae that ultimately result in hyphal death [30,31]. But the endofungal lifestyle of this bacterial species has not been reported earlier.

We have tested the efficiency of the bacterial isolate to migrate and invade the hyphae of different fungal strains belonging to different taxonomic groups. We observed that, the bacterial isolate was unable to migrate along the hyphae of the tested Ascomycete and Basidiomycete (except *Fusarium oxysporum*). On the other hand, the bacterium was able to invade and migrate along the hyphae of *Mucor irregularis*, a member of Zygomycete. It is well known that Zygomycete fungal cell wall is composed of a special type of polysaccharide chitosan [45], which is a derivative of chitin, the general polysaccharide found in Ascomycete and Basidiomycete cell walls. A study on endosymbiotic bacteria *Burkholderia rhizoxinica* revealed that a type II secretion system was responsible for release of chinolytic enzymes [23]. Release of chitinolytic enzymes and zygomycete cell wall binding factors allow the partial degradation of the fungal cell wall.

Our bacterial isolate was unable to spread along the hyphae of the tested Ascomycota fungi (except *Fusarium oxysporum*). *Serratia marcescens* was previously reported to spread over Zygomycete *Rhizopus oryzae*, but not over Ascomycete *(Aspergillus* sp.) in a chitinase independent manner [31]. Some adhesion factors are thought to be responsible for attachment to the fungal cell wall. The physical contact must occur between the bacterium and the fungus that may be dependent upon these factors [46]. Few fungal species/phyla may also produce surface or secreted defense factors that interfere bacterial adhesions [31]. Bacterial attachment and invasion may occur only in those fungi that possess these adhesion factors on the surface of the hyphae or lack the defense factors. In our experiments, we observed that *Serratia* was able to migrate through the fungal hyphae aerially, only after coming in contact with the fungal hyphae. No sign of chemotaxis was detected for the bacterial isolate indicating the contact dependent binding of the bacteria to the host hyphal surface.

In contrast to the other Ascomycete, *Fusarium oxysporum* allowed the spreading of the bacterium over its mycelia. The bacterium was able to spread, invade and kill the *F. oxysporum* hyphae on solid medium. The viability of infected hyphae was lost after infection with the bacterium, suggesting strong antagonistic activity of the bacterial isolate against *Fusarium oxysporum*.

We have isolated a pink-red pigment prodigiosin, which has been reported to show several bioactive properties including antibacterial [47], anticancer [48], immunosuppressive [49], insecticidal [50] and larvicidal [51] activities. In this study, we have studied the antagonistic potential of prodigiosin against zygomycete and ascomycete fungi. Prodigiosin was able to suppress the growth of fungal strains, indicating the noxius effect of prodigiosin against fungi.

Due to the increase in cell membrane permeability, propidium iodide could enter into the fungal hyphae that was detected as red fluorescence. Increasing concentrations of prodigiosin enhanced the membrane permeability denoting a positive correlation with the invasion of *S. marcescens* into *M. irregularis* hyphae.

Quantitative real-time PCR revealed same level of expression for all the tested prodigiosin biosynthetic genes in both control and interaction cultures. As prodigiosin was produced in both bacterial pure culture and bacterial fungal co-culture, it indicated that prodigiosin production is not dependent on the interaction process. Although, it may have crucial role in the invasion process through pore formation in the plasma membrane. Involvement of prodigiosin along with chitinases were previously reported during antagonistic activity of *Serratia marcescens* against fungal pathogens [52]. Chitinase and other chitinolytic enzymes secreted by type II secretion system were reported to be involved in the invasion of fungal hyphae by *Burkholderia* sp. [23]. In our study, the expression level of chitinase A was using qRT-PCR. The results indicated that there was very low expression level of chitinase A in both control and interaction cultures. These results confirmed that the interaction process is chitinase A independent. Chitinase A independent killing mechanism of *Serratia marcescens* was described by Hover and co-workers [31], where the bacterial cells bound and spread over the hyphal surface by biofilm formation. However, we do not deny the involvement of other chitinases of *S. marcescens* in the interaction process, which were not tested in the present study.

We have studied the involvement of other secretion machineries in the fungal invasion by *S. marcescens* D1. Comparative genome analysis of *Serratia marcescens* revealed that the type VI secretion system is abundant among various strains [32], which is used to target other bacterial competitors [33]. The T6SS effectors to antagonize fungal pathogens has also been reported in *S. marcescens* [53]. In our study, the expression of transcripts for TssJ, an outer membrane protein involved in the T6SS assembly, and an outer membrane murein-lipoprotein were tested for their relative expression during bacterial pure culture and bacterial-fungal interaction culture. Significant up-regulation of these two proteins indicated the involvement of T6SS in the invasion process. Mutation based approaches are needed to elucidate the actual role of T6SS in the invasion process.

## Conclusion

Based on our findings, the interaction between *Serratia marcescens* D1 and *Mucor irregularis* SS7 could be regarded as a balanced antagonism, where the endofungal bacterium live inside the fungal hyphae without showing lethal effect. This study may be considered as the first report to demonstrate the endofungal association of *S. marcescens*. Further study on the molecular mechanisms employed by its fungal partner may reveal the unsolved queries why and how this fungus is allowing the bacterium to live inside its hyphae.

## Supporting information

**S1 Fig. PCR based screening of the presence of bacterial 16S rRNA gene in the total genomic DNA isolated from the fungal cultures. A**. Amplification of 16S rRNA gene in the screened cultures. Lane 1: 1 kb DNA ladder; Lane 2–8: sample IDs SS7, OR4.1, AAU-R4, AAU-R6, SC2.2, SC4.6 and HB8, respectively; Lane 9: sample SS1 showing no amplification for 16S rRNA gene; Lane 10: *Escherichia coli* K12 as positive control for amplification. **B**. Amplification of 16S rRNA gene from the total genomic DNA isolated after two subsequent subcultures of the positive isolates. Lane 1: 1 kb DNA ladder; Lane 2–8: sample IDs SS7, OR4.1, AAU-R4, AAU-R6, SC2.2, SC4.6 and HB8, respectively; Lane 9: *Escherichia coli* K12 as

positive control for amplification.
(DOCX)

**S2 Fig. FAME analysis report showing similarity index of isolate D1.**
(DOCX)

**S3 Fig. Microscopic observations of the fungal hyphae and spores of the fungal isolate SS7.**
**A**. hyphae of the fungal isolate showing rhizoids; **B**. A young sporangia on the terminal
hyphae; **C**. spores of the fungal isolates; scale bar = 10 μm.
(DOCX)

**S4 Fig. Interaction of *Serratia marcescens* with Ascomycetes fungi.** *Mucor irregularis* SS7
was used as positive control. Photographs were taken after 48 h of bacterial interaction with
the fungal hyphae. Pink red pigmentation indicated bacterial spreading.
(DOCX)

**S5 Fig. Interaction of *Serratia marcescens* with Basidiomycetes fungi.** Photographs were
taken after 48 h of bacterial interaction with the fungal hyphae. Pink red pigmentation indi-
cated bacterial spreading.
(DOCX)

**S6 Fig. *M. irregularis* SS7 spore count at different time interval after treatment with *S.***
***marcescens* D1 and its pink-red pigment prodigiosin.** Error bars represents the standard
deviations of three independent replications.
(DOCX)

**S7 Fig. Pore formation on cell membrane of *Mucor irregularis* SS7 detected by the intensity**
**of propidium iodide with increasing concentrations of prodigiosin.** Upper rows contains
the brighfield images of the fungal hyphae. Lower row shows the images with red fluorescence
of propidium iodide.
(DOCX)

**S1 Table. Primers designed for quantitative real-time PCR analysis.**
(DOCX)

**S2 Table. Morphological and biochemical test results for the bacterial isolate D1.**
(DOCX)

**S3 Table. Results of the antibiotic susceptibility test for the bacterial isolate D1.**
(DOCX)

**S4 Table. Inhibitory effect of prodigiosin on fungal cultures.**
(DOCX)

## Acknowledgments

Authors are grateful to Department of Agricultural Biotechnology and DBT-North East Centre
for Agricultural Biotechnology (formerly DBT-AAU Centre), Assam Agricultural University
for providing laboratory space and required facilities for carrying out the study.

## Author Contributions

**Conceptualization:** Dibya Jyoti Hazarika, Robin Chandra Boro.

**Data curation:** Dibya Jyoti Hazarika, Trishnamoni Gautom.

**Formal analysis:** Dibya Jyoti Hazarika, Trishnamoni Gautom, Assma Parveen, Gunajit Goswami.

**Funding acquisition:** Robin Chandra Boro.

**Investigation:** Dibya Jyoti Hazarika, Trishnamoni Gautom, Assma Parveen, Robin Chandra Boro.

**Methodology:** Dibya Jyoti Hazarika, Gunajit Goswami, Madhumita Barooah.

**Project administration:** Mahendra Kumar Modi, Robin Chandra Boro.

**Resources:** Madhumita Barooah, Mahendra Kumar Modi, Robin Chandra Boro.

**Supervision:** Madhumita Barooah, Mahendra Kumar Modi, Robin Chandra Boro.

**Validation:** Dibya Jyoti Hazarika, Trishnamoni Gautom, Gunajit Goswami, Robin Chandra Boro.

**Visualization:** Dibya Jyoti Hazarika.

**Writing – original draft:** Dibya Jyoti Hazarika.

**Writing – review & editing:** Dibya Jyoti Hazarika, Trishnamoni Gautom, Assma Parveen, Gunajit Goswami, Madhumita Barooah, Mahendra Kumar Modi, Robin Chandra Boro.

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
