## [Decision Letter · Decision Letter 0]

13 Jan 2020

PONE-D-19-27514

Mechanism of interaction of an endofungal bacterium Serratia marcescens D1 with its host and non-host fungi

PLOS ONE

Dear Dr. Boro,

Thank you for submitting your manuscript to PLOS ONE. After careful consideration, we feel that it has merit but does not fully meet PLOS ONE’s publication criteria as it currently stands. Therefore, we invite you to submit a revised version of the manuscript that addresses the points raised during the review process.

We would appreciate receiving your revised manuscript by Feb 27 2020 11:59PM. To enhance the reproducibility of your results, we recommend that if applicable you deposit your laboratory protocols in protocols.io, where a protocol can be assigned its own identifier (DOI) such that it can be cited independently in the future. For instructions see: http://journals.plos.org/plosone/s/submission-guidelines#loc-laboratory-protocols

We look forward to receiving your revised manuscript.

Kind regards,

Pankaj Kumar Arora

Academic Editor

PLOS ONE

Journal Requirements:

2. Our internal editors have looked over your manuscript and determined that it is within the scope of our Microbes & Host Cell Membrane Interactions Call for Papers. This collection of papers is headed by a team of Guest Editors for PLOS ONE: Nihal Altan-Bonnet, Stacey Gilk, Richard Hayward, and Luis Schang. The Collection will encompass a diverse range of research articles that contribute to our understanding of the mechanisms through which viruses, bacteriophage, bacteria, fungi, parasites, and microbial toxins interact with host cell and host-derived membranes..  Additional information can be found on our announcement page: https://collections.plos.org/s/microbes.

If you would like your manuscript to be considered for this collection, please let us know in your cover letter and we will ensure that your paper is treated as if you were responding to this call. If you would prefer to remove your manuscript from collection consideration, please specify this in the cover letter.

Reviewers' comments:

Reviewer's Responses to Questions

**Comments to the Author**

1. Is the manuscript technically sound, and do the data support the conclusions?

Reviewer #1: Yes

2. Has the statistical analysis been performed appropriately and rigorously? 

Reviewer #1: Yes

3. Have the authors made all data underlying the findings in their manuscript fully available?

Reviewer #1: Yes

4. Is the manuscript presented in an intelligible fashion and written in standard English?

Reviewer #1: Yes

5. Review Comments to the Author

Reviewer #1: This manuscript well written and clearly presented. Author did characterization of an endofungal bacterium Serratia marcescens D1 from Mucor irregularis SS7 hyphae. Upon re-inoculation, colonization of the endobacterium S. marcescens D1 in the hyphae of Mucor irregularis SS7 was demonstrated using stereo microscopy. However, S. marcescens D1 failed to invade into the hyphae of the tested Ascomycetes (except Fusarium oxysporum) and Basidiomycetes. Remarkably, Serratia marcescens D1 could invade and spread over the culture of F. oxysporum that resulted in mycelial death.

Minor comments:

Please plot a phylogenetic tree for strain Serratia marcescens D1 with it closest strains. mention about 16S rDNA distance value in MS.

6. PLOS authors have the option to publish the peer review history of their article (what does this mean?). If published, this will include your full peer review and any attached files.

Reviewer #1: Yes: Ramprasad EVV

---

## [Author Response · Author response to Decision Letter 0]

23 Jan 2020

#Reviewer 1:

Thank you for your critical examination of the manuscript. A track changed revised manuscript is included.

As per your suggestion, the phylogenetic trees have been included to the revised manuscript (Figure 3). Methodologies for tree building as well as results of the phylogenetic analysis of both bacterial and fungal isolates have been described in the revised manuscript.

Additionally, Figure numbers have been changed and renamed. A few grammatical mistakes have been corrected in the revised manuscript.

---

## [Decision Letter · Decision Letter 1]

9 Mar 2020

Mechanism of interaction of an endofungal bacterium Serratia marcescens D1 with its host and non-host fungi

PONE-D-19-27514R1

Dear Dr. Boro,

We are pleased to inform you that your manuscript has been judged scientifically suitable for publication and will be formally accepted for publication once it complies with all outstanding technical requirements.

With kind regards,

Pankaj Kumar Arora

Academic Editor

PLOS ONE

Additional Editor Comments (optional):

Reviewers' comments:

Reviewer's Responses to Questions

**Comments to the Author**

1. If the authors have adequately addressed your comments raised in a previous round of review and you feel that this manuscript is now acceptable for publication, you may indicate that here to bypass the “Comments to the Author” section, enter your conflict of interest statement in the “Confidential to Editor” section, and submit your "Accept" recommendation.

Reviewer #1: All comments have been addressed

2. Is the manuscript technically sound, and do the data support the conclusions?

Reviewer #1: Yes

3. Has the statistical analysis been performed appropriately and rigorously? 

Reviewer #1: N/A

4. Have the authors made all data underlying the findings in their manuscript fully available?

Reviewer #1: Yes

5. Is the manuscript presented in an intelligible fashion and written in standard English?

Reviewer #1: Yes

6. Review Comments to the Author

Reviewer #1: (No Response)

7. PLOS authors have the option to publish the peer review history of their article (what does this mean?). If published, this will include your full peer review and any attached files.

Reviewer #1: Yes: Dr Ramprasad EVV

---

## [Editor Report · Acceptance letter]

10 Apr 2020

PONE-D-19-27514R1 

Mechanism of interaction of an endofungal bacterium *Serratia marcescens* D1 with its host and non-host fungi 

Dear Dr. Boro:

I am pleased to inform you that your manuscript has been deemed suitable for publication in PLOS ONE. Congratulations! Your manuscript is now with our production department. 

With kind regards,

on behalf of

Dr. Pankaj Kumar Arora 

Academic Editor

PLOS ONE